# Utilizing Image Transforms and Diffusion Models for Generative Modeling of Short and Long Time Series

**Ilan Naiman*  Nimrod Berman*  Itai Pemper  Idan Arbiv  Gal Fadlon  Omri Azencot**
Department of Computer Science
Ben-Gurion University of The Negev
{naimani, bermann, itaipem, arbivid, galfad}@post.bgu.ac.il
azencot@cs.bgu.ac.il

## Abstract

Lately, there has been a surge in interest surrounding generative modeling of time series data. Most existing approaches are designed either to process short sequences or to handle long-range sequences. This dichotomy can be attributed to gradient issues with recurrent networks, computational costs associated with transformers, and limited expressiveness of state space models. Towards a unified generative model for varying-length time series, we propose in this work to transform sequences into images. By employing invertible transforms such as the delay embedding and the short-time Fourier transform, we unlock three main advantages: i) We can exploit advanced diffusion vision models; ii) We can remarkably process short- and long-range inputs within the same framework; and iii) We can harness recent and established tools proposed in the time series to image literature. We validate the effectiveness of our method through a comprehensive evaluation across multiple tasks, including unconditional generation, interpolation, and extrapolation. We show that our approach achieves consistently state-of-the-art results against strong baselines. In the unconditional generation tasks, we show remarkable mean improvements of $58.17\%$ over previous diffusion models in the short discriminative score and $132.61\%$ in the (ultra-)long classification scores. Code is at `https://github.com/azencot-group/ImagenTime`.

## 1   Introduction

Generative modeling of real-world information such as images [72], texts [13], and other types of data [99, 55, 8] has drawn increased attention recently. In this work, we focus on the setting of generative modeling (GM) of general time series information. There are several factors that govern the complexity required from sequential data generators including the sequence length, its number of features, the appearance of transient vs. long-range effects, and more. Existing generative models for time series are typically designed either for multivariate short-term sequences [44, 19] or univariate long-range data [103], often resulting in separate and completely different neural network frameworks. However, a natural question arises: Can one develop a unified framework equipped to handle both high-dimensional short sequences and low-dimensional long time series?

Earlier approaches for processing time series based on recurrent neural networks (RNNs) handled short sequences well [62, 3, 43, 76], however, modeling long-range dependencies turned out to be significantly more challenging. Particularly, RNNs suffer from the well-known vanishing and exploding gradient problem [9, 70] that prevents them from learning complex patterns and long-range dependencies. To address long-context modeling and memory retention, extensive research is devoted to approaches such as long short-term memory (LSTM) models [42], unitary evolution RNNs [5]

---

*Equal Contribution

38th Conference on Neural Information Processing Systems (NeurIPS 2024).

and Lipschitz RNNs [24]. A different approach for processing sequential information is based on the Transformer [93], eliminating any recurrent connections. Recent remarkable results have been obtained with transformers on natural language processing [13] and time series forecasting [96, 104, 68] tasks. Alas, transformers are underexplored as generative models for long-range time series data. This may be in part due to their computational costs that scale quadratically as $\mathcal{O}(L^2)$ with the sequence length $L$, and in part because transformer forecasters are inferior to linear tools [101].

Beyond RNNs and the Transformer, recent works have considered the state space model (SSM) for modeling long-range time series information. For instance, the structured SSM (S4) [36] employed a parameterization that reduced computational costs via evaluations of Cauchy kernels. Further, the deep linear recurrent unit (LRU) is inspired by the similarities between SSMs and RNNs, and it demonstrated impressive performance in modeling long-range dependencies (LRD). Still, generative modeling of long-range sequential data via state space models remains largely underexplored. Recent work suggested LS4 [103], a latent time series generative model that builds upon linear state space equations. LS4 utilizes autoregressive dependencies to expressively model time series (potentially non-stationary) distributions. However, this model struggles with short-length sequences as we show in our study, potentially due to limited expressivity of linear SSMs.

To overcome gradient issues of recurrent backbones, temporal computational costs of transformers, and expressivity problems of SSMs, we represent time series information via small-sized *images*. Transforming raw sequences to other encodings has been useful for processing audio [34] as well as general time series data [95, 38, 56]. Moreover, a similar approach was employed to generative modeling of time series with generative adversarial networks (GANs) [12, 39]. However, unstable training dynamics and mode collapse negatively affect the performance of GAN-based tools [59]. In contrast, transforming time series to images is underexplored in the context of generative *diffusion* models. There are several fundamental advantages to our approach. First, there have been remarkable advancements in diffusion models for vision data that we can exploit [81, 40, 86, 45]. Second, using images instead of sequences elegantly avoids the challenges of long-term modeling. For instance, a moderately-sized $256 \times 256$ image corresponds to a time series of length up to $65k$, as we show in Sec. 3. Finally, there is a growing body of literature dealing with time series as images on generative, classification, and forecasting tasks, whose results can be applied in our work and in future studies.

In this work, we propose a new diffusion-based framework for generative modeling of general time series data, designed to seamlessly process both short-, long-, and *ultra*-long-range sequences. To evaluate our method, we consider standard benchmarks for short to ultra-long time series focusing on unconditional generation. Our approach supports efficient sampling, and it attains state-of-the-art results in comparison to recent generative models for sequential information. As far as we know, there are no existing tools handling both short and long sequence data. In addition to its strong unconditional generation capabilities, our approach is also tested in conditional scenarios involving the interpolation of missing information and extrapolation. Overall, we obtained state-of-the-art results in such cases with respect to existing tools. We further analyze and ablate our technique to motivate some of our design choices. The contributions of our work can be summarized as follows:

1. We view generative modeling of time series as a visual challenge, allowing to harness advances in time series to image transforms as well as vision diffusion models.

2. We develop a novel generative model for time series that scales from short to very long sequence lengths without significant modifications to the neural architecture or training method.

3. Our approach achieves state-of-the-art results in comparison to strong baselines in unconditional and conditional generative benchmarks for time series of lengths in the range $[24, 17.5k]$. Particularly, we attain the best scores on a new challenging benchmark of very long sequences that we introduce.

## 2 Related work

**Time series to image works.** Motivated by the success of convolutional neural networks on vision data, several works have transformed time series to images using Gramian Angular Fields [95], Recurrence Plots [38], and Line Graphs [56]. This innovation allows leveraging computer vision techniques, tested on tasks such as time series classification and imputation. In speech analysis and processing, the short-time Fourier transform (STFT) stands out as a widely used method [1, 2, 94, 26]. It tracks the changes in frequency components over time, making it essential for analyzing and understanding audio and speech data. Recent research [71, 17] has explored mel-spectrogram

transforms within diffusion models, including integration with advanced latent diffusion spaces [58]. Furthermore, combining time series images and Wasserstein GANs [12, 39] have been considered for generative modeling. Yet, representing general time series as images within diffusion models for tasks such as unconditional generation, interpolation, and extrapolation, remains largely underexplored. The goal of this work is to make a step toward bridging this gap.

**Diffusion models.** Both denoising diffusion probabilistic models (DDPM) [81, 40] and score-based generative models [84, 85] have demonstrated their effectiveness across diverse domains including images [74, 41], audio [15, 52], and graphs [69, 97, 10]. Song et al. [86] showed that DDPM and score-based models can be both interpreted as stochastic differential equations (SDE). Further works focused on improving generation quality by using latent diffusion processes in autoencoder architectures [74]. Another research direction deals with lowering the number of neural function evaluations (NFEs), which originally ranged from hundreds to thousands NFEs. For instance, Karras et al. [45] obtain a low Fréchet inception distance (FID) with only 35 evaluations, whereas the recent consistency models [82] achieved comparable results with only a single function evaluation.

**Generative modeling of time series.** Generative adversarial networks (GANs) [31] have shown remarkable success in generating realistic data across various domains. Specifically, their application to time series information by joint optimization of supervised and adversarial objectives via TimeGAN captured the inherent dynamics of real-world signals [99]. Similarly, GT-GAN [44] utilizes diverse tools including ordinary differential equations (DE) [16], neural controlled DE [51], and continuous time-flow processes to model both regularly- and irregularly-sampled data [21]. Nevertheless, GANs suffer from challenges, primarily due to unstable training dynamics and mode collapse [59]. Beyond GANs, variational autoencoders (VAEs) have been also considered for generative modeling of sequential data [22, 54, 73], where the work [66] achieved strong results using Koopman-based approaches [7, 6, 65, 11]. To process long-range dependencies and stiff dynamics [79], Zhou et al. [103] introduced LS4, a latent generative model based on linear state space equations. Following the success of diffusion models in other domains, there is a growing desire to adapt them for time series data. However, this adaption is not straightforward and entails the design of a suitable backbone [88, 57, 19, 100, 67]. Other approaches focused on regression problems [49], based on manifold learning tools [47, 48]. Instead, we propose a new framework for generative modeling of time series by transforming such data to images and using existing strong diffusion vision models.

# 3 Background

In what follows, we state the problem, we mention two effective time series to image transformations, and we briefly discuss the essentials of diffusion-based generative modeling.

**Problem statement.** We address the problem of generating time series (TS), sampled from a learned distribution $\tilde{p}(x)$ that is similar to an unknown distribution $p(x)$, for which we have a set of observed TS data. The given observations include data samples $x \in \mathbb{R}^{L \times K}$, where $L$ represents the sequence length and $K$ denotes the number of features. Formally, the generative modeling task is often termed "unconditional generation" [37], and it entails learning a model $M$ capable of sampling unseen time series $\tilde{x}$ from $\tilde{p}(x)$. Additionally, our work addresses a secondary problem known as "conditional generation". In this setting, given an additional signal $c$, we learn the unknown (conditional) distribution $p(x|c)$. For example, the signal $c$ can be an observed part from the TS. This conditional modeling proves useful for tasks such as time series interpolation and extrapolation.

**Time series to image transforms.** We focus in our study on two invertible time series to image transformations: 1) the delay embedding; and 2) the short time Fourier transform. We provide below a brief overview of these transforms and their inverse. We consider additional transforms and we discuss more details in App. A. Fig. 3 illustrates a time series signal and its related images.

*Delay embeddings* [87] transform a univariate time series $x_{1:L} \in \mathbb{R}^L$ to an image by arranging the information of the series in columns and pad if needed. Let $m, n$ be two user parameters representing

the skip value and the column dimension, respectively. We construct the following matrix $X$,

$$X = \begin{bmatrix} x_1 & x_{m+1} & \cdots & x_{L-n} \\ \vdots & \vdots & \cdots & \vdots \\ x_n & x_{n+m+1} & \cdots & x_L \end{bmatrix} \in \mathbb{R}^{n \times q} \,, \tag{1}$$

where $q = \lceil (L-n)/m \rceil$. The image $x_{\text{img}}$ is created by padding with zeros to fit the neural network input constraints. Given $x_{\text{img}}$, the original time series $x_{1:L}$ can be extracted in multiple ways. For instance, if $m = 1$, then $x_{1:L}$ is formed by concatenating the first row and last column of $x_{\text{img}}$. The delay embedding scales naturally to long sequences, e.g., setting $m = n = 256$ allows to encode $65k$ sequences with $256 \times 256$ images.

*Short Time Fourier Transform (STFT)* [35] is a well-known transformation that maps a signal from its original domain into the frequency domain. To preserve the temporal structure, STFT applies a rolling window on the time axis, extracting time series segments for which the fast Fourier transform (FFT) is applied. Given an input signal $x \in \mathbb{R}^{L \times K}$, STFT produces an image $x_{\text{img}} \in \mathbb{R}^{2K \times H \times W}$, where the channels are doubled to store the real and imaginary parts, and $H, W$ are derivatives of user parameters. STFT requires a minimum window length, and thus, short sequences may require a linear interpolation to match length constraints. Remarkably, the short time Fourier transform is invertible via reverse STFT with a negligible loss of information. Importantly, in contrast to the common practice in audio processing, we do not further compute the spectrogram of STFT, avoiding non-trivial inverse transformations.

**Diffusion models.** Diffusion processes gradually add noise to an image, following a predefined noise scheduling scheme. Generating new images is possible by learning a model that removes noise. The diffusion process $\{\mathbf{x}(t)\}_{t=0}^T$ is the path of a stochastic differential equation (SDE) [86], where an initial sample $\mathbf{x}(0)$ is drawn from the data distribution $p_0(\mathbf{x})$. The initial sample is modified to $\mathbf{x}(T)$, sampled from a simple prior distribution such as a normal Gaussian $\mathcal{N}(0, I)$. Formally, the forward process is governed by an SDE of the form,

$$d\mathbf{x} = f(\mathbf{x}, t)dt + g dw \,, \tag{2}$$

where $f(\cdot, t) : \mathbb{R}^d \to \mathbb{R}^d$ represents the drift coefficient, $g \in \mathbb{R}$ is the diffusion scalar, and $w$ denotes a standard Wiener process. To facilitate sampling, we need to derive the reverse SDE. It is well-known that the reverse process [4] is given by,

$$d\mathbf{x} = [f(\mathbf{x}, t) - g^2 \nabla_{\mathbf{x}} \log p_t(\mathbf{x})]d\bar{t} + g d\bar{w} \,, \tag{3}$$

where $\bar{t}$ denotes reverse time and $\bar{w}$ is a reverse Wiener process. Given Eq. (3), one can derive a deterministic process, characterized by trajectories that share identical marginal probability densities. Formally, we obtain the following ordinary differential equation (ODE),

$$d\mathbf{x} = f(\mathbf{x}, t) - g^2 \nabla_{\mathbf{x}} \log p_t(\mathbf{x}) \,. \tag{4}$$

Diffusion models compute an estimator $s_\theta(\mathbf{x}, t)$ to approximate the infeasible $\nabla_{\mathbf{x}} \log p_t(\mathbf{x})$ via

$$\min_\theta \mathbb{E}_t \{\mathbb{E}_{x_0, x_t} |s_\theta(\mathbf{x}, t) - \nabla_{\mathbf{x}} \log p_{0t}(\mathbf{x}_t | x_0)|_2^2\} \,, \tag{5}$$

where $p_{0t}$ denotes the joint distribution of the initial data and noisy sample. In practice, minimizing Eq. (5) is done by learning the noise pattern of input images. For a more comprehensive discussion regarding score-based models, we refer to [86, 45]. We specify in Sec. 4 the particular diffusion model employed in this work, along with further additional details.

## 4 Method

Our approach to generative modeling of time series information is based on the following simple observation and straightforward idea. We observe that diffusion models for vision have demonstrated remarkable progress and results recently [40, 74, 72]. Therefore, our idea is to transform sequences to images, allowing their processing using established diffusion vision models. Fortunately, there are several efficient time series to image maps with effective inverse transforms [87, 35, 95, 56]. Our computational pipeline is composed of three main building blocks: 1) a time series to image module; and 2) a diffusion model; and 3) an image to time series component. The diffusion model is the

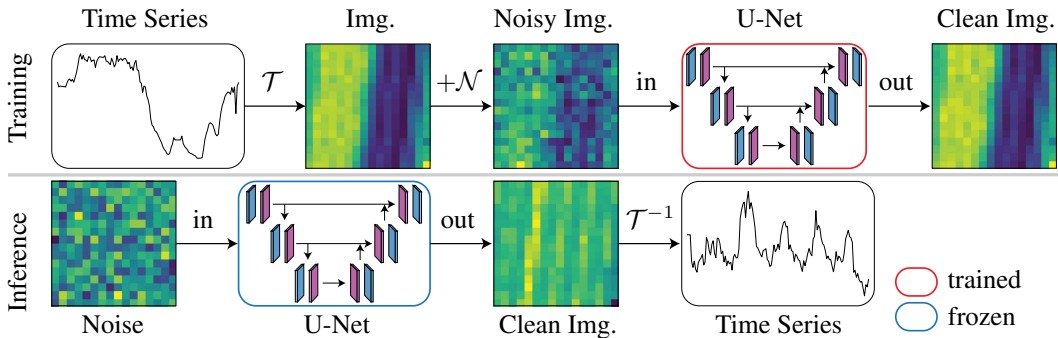

Figure 1: Our training pipeline (top) involves transforming a time series signal to its e.g., delay embedding image, process the image with a diffusion model, and output its cleaned version. During inference (bottom), we sample from a standard normal distribution and obtain a clean image using the trained diffusion model. Finally, we transform the image back to the time series domain.

only learnable parameter-based part of our neural network. We illustrate our generative modeling framework for time series information in Fig. 1, depicting the above building blocks with proper notations for inputs and outputs. Formally, given an input time series $x \in \mathbb{R}^{L \times K}$ with $L$ the sequence length and $K$ the number of features, we transform it to an image $x_{\text{img}} \in \mathbb{R}^{C \times H \times W}$. Noise is added to the latter image yielding the tensor $x_{\text{img}}(t)$ which is processed with our diffusion model, whose output $s(x_{\text{img}}, t) \in \mathbb{R}^{C \times H \times W}$ represents the cleaned image. During inference, noise $x_{\text{img}}(T)$ is sampled from $\mathcal{N}(0, I)$, iterated backward to $x_{\text{img}}(0)$ and transformed to a time series $\tilde{x} \in \mathbb{R}^{L \times K}$.

There are several options to choose from regarding the time series to image (`ts2img`) transform, $\mathcal{T} : \mathbb{R}^{L \times K} \rightarrow \mathbb{R}^{C \times H \times W}$. While all transforms are applicable in our framework, we opt for `ts2img` maps that are efficient to compute, provide informative images, scalable across short and long sequences, and have a closed-form inverse. For instance, line graphs [56] are efficient with a closed-form inverse, however, they produce images that are mostly non-informative as they contain blank pixels. Similarly, the Gramian angular field transform [95] essentially stores the sequence in its main diagonal, and thus, it is not straightforward to apply it to long-range data. In this work, we focus on using the delay embedding and short time Fourier transforms. Both `ts2img` maps satisfy all requirements above. Moreover, our empirical ablation analysis in Sec. 5.5 highlights that these transformations attain the best results on average. The inverse transforms $\mathcal{T}^{-1}$ for delay embedding and STFT are parameter-less, deterministic, and highly efficient. See Sec. 3 and App. A.

At the heart of our **ImagenTime** framework lies the generative diffusion model backbone. Diffusion models for vision data have enjoyed increased attention over the past few years, with strong techniques appearing at an unprecedented rate [81, 84, 40, 85, 86, 74]. One limitation, shared among all diffusion models, is the requirement to iteratively denoise the image during inference, resulting in costly neural function evaluations (NFEs). While multiple works focused on alleviating this issue [77, 83, 28, 82], the work by Karras et al. [45] offers an enhanced score-based model with a good balance between rapid sampling and high-quality generations. Specifically, they presented a clearer design space for the factors that determine the performance of diffusion models, and they suggested EDM that employs a second-order ODE for the reverse process, yielding low FID images in 35 NFEs. Thus, we utilize in this work the EDM diffusion model as our generative backbone.

We conclude by briefly discussing the training and inference procedures. During *training*, we process batches of time series data $X$, for which we apply $\mathcal{T}$ using either delay embedding or STFT to obtain a batch of images, i.e., $X_{\text{img}} = \mathcal{T}(X)$. Subsequently, we employ the training procedure of EDM to learn the score function $s_\theta(X_{\text{img}}, t)$. For *inference*, we use the trained EDM model and we compute the reverse ODE in Eq. (4) for sampling new data points. In practice, we follow the same inference procedure specified in [45]. Finally, given a batch of sampled images, $\tilde{X}_{\text{img}}$, we apply the inverse transform $\mathcal{T}^{-1}$ to achieve a batch of generated time series samples, i.e., $\tilde{X} = \mathcal{T}^{-1}(\tilde{X}_{\text{img}})$.

## 5 Experiments

We use standard unconditional and conditional quantitative and qualitative benchmarks to extensively validate our framework's ability to generate high-quality time series samples. First, we test our

Table 1: Error measures for the short time series unconditional discriminative and prediction tasks.

| Method | Stocks disc↓ | Stocks pred↓ | Energy disc↓ | Energy pred↓ | MuJoCo disc↓ | MuJoCo pred↓ |
|---|---|---|---|---|---|---|
| KoVAE | **.009 ± .006** | .037 ± .000 | .143 ± .011 | .251 ± .000 | .076 ± .017 | .038 ± .002 |
| DiffTime | .050 ± .017 | .038 ± .001 | .101 ± .019 | .250 ± .003 | .059 ± .009 | .042 ± .000 |
| GT-GAN | .077 ± .031 | .040 ± .000 | .221 ± .068 | .312 ± .002 | .245 ± .029 | .055 ± .000 |
| TimeGAN | .102 ± .021 | .038 ± .001 | .236 ± .012 | .273 ± .004 | .409 ± .028 | .082 ± .006 |
| RCGAN | .196 ± .027 | .040 ± .001 | .336 ± .017 | .292 ± .004 | .436 ± .012 | .081 ± .003 |
| C-RNN-GAN | .399 ± .028 | .038 ± .000 | .449 ± .001 | .483 ± .005 | .412 ± .095 | .055 ± .004 |
| T-Forcing | .226 ± .035 | .038 ± .001 | .483 ± .004 | .315 ± .005 | .499 ± .000 | .142 ± .014 |
| P-Forcing | .257 ± .026 | .043 ± .001 | .412 ± .006 | .303 ± .005 | .500 ± .000 | .102 ± .013 |
| WaveNet | .232 ± .028 | .042 ± .001 | .397 ± .010 | .311 ± .006 | .385 ± .025 | .333 ± .004 |
| WaveGAN | .217 ± .022 | .041 ± .001 | .363 ± .012 | .307 ± .007 | .357 ± .017 | .324 ± .006 |
| LS4 | .199 ± .065 | .068 ± .013 | .474 ± .003 | .251 ± .000 | .333 ± .029 | .062 ± .006 |
| Ours | .037 ± .006 | **.036 ± .000** | **.040 ± .004** | **.250 ± .000** | **.007 ± .005** | **.033 ± .001** |

framework on short-term and long-term standard time series unconditional generation benchmarks (Sec. 5.1, Sec. 5.2). Then, we introduce a novel benchmark for ultra-long sequences (above 10k steps) and evaluate our method in comparison to strong baselines (Sec. 5.3). Then, we consider interpolation and extrapolation benchmarks, similar to [78], to test our model on conditional generation tasks (Sec. 5.4). Further, we extended these benchmarks with additional short- and ultra-long setups, which test the framework's robustness to lengths. Finally, we conclude with an extensive ablation of our framework (Sec. 5.5). More details on the experimental settings can be found in App. B.

## 5.1 Short-Term Unconditional Generation

**Data, baselines, and metrics.** We employ our framework on the unconditional generation benchmark reported in [19]. The benchmark includes four synthetic and real-world datasets with a fixed length of 24. The first dataset, *Stocks*, consists of daily historical Google stock data from 2004 to 2019, comprising six channels: high, low, opening, closing, and adjusted closing prices, as well as volume. This data lacks periodicity and is dominated by random walks. The second dataset, *Energy*, is a multivariate appliance energy prediction dataset [14], featuring 28 channels with correlated features, and it exhibits noisy periodicity and continuous-valued measurements. The third dataset, *MuJoCo* (Multi-Joint dynamics with Contact), serves as a versatile physics generator for simulating TS data with 14 channels [89]. We report results on the simple synthetic *Sine* dataset of sine functions in App. C.1. Our framework is compared with state-of-the-art short-term time series generative models. KoVAE [66], DiffTime [19], GT-GAN [44], TimeGan [99], RCGAN [25], C-RNN-GAN [64], T-Forcing [33], P-forcing [32], WaveNet [91], WaveGAN [23], and LS4 [103], which is the state-of-the-art generative model for modeling long sequences. The benchmark employs two metrics: 1) The *Predictive (pred)* metric assesses the utility of the generated data. 2) The *Discriminative (disc)* metrics gauge the similarity of distributions using a proxy discriminator. For all experiments, we used the *delay embedding* transform with an embedding of $n = 8$ and a delay of $m = 3$, yielding a $8 \times 8$ image. We use 18 sampling steps with the EDM model [45] as the diffusion generative backbone.

**Quantitative and qualitative results.** The results for the short-term unconditional benchmark are shown in Tab. 1. Our framework achieves state-of-the-art results on all datasets and metrics. Particularly, we note MuJoCo, where we improved the second-best method by 88% and 21% in the discriminative and predictive scores. In general, the second-best approach is DiffTime. Importantly, while LS4 performs well on long sequences, our results indicate that it struggles with short sequences. In comparison, we will show below that in addition to obtaining SOTA results on short-term time series, we also achieve strong results in the long-term case (Sec. 5.2). We also evaluate our method using two common qualitative tests [99]. First, we compute a two-dimensional t-SNE [92] embedding for real and synthetic data. The desired outcome is that both datasets span similar regions and shapes in 2D. We plot the embeddings of the real data, ours and GT-GAN in Fig. 2A, highlighting that our generated point clouds are closer to the real data in comparison GT-GAN. Tab. 11 reports the Wasserstein distances between the t-SNE embeddings of the generated data and the real data, showing that our approach is superior to GT-GAN. Second, we estimate the probability density functions in Fig. 2D. Our approach generates densities similar to the real densities, whereas GT-GAN introduces noticeable errors. The rest of the short-term datasets' qualitative analysis appears in App. C.3.

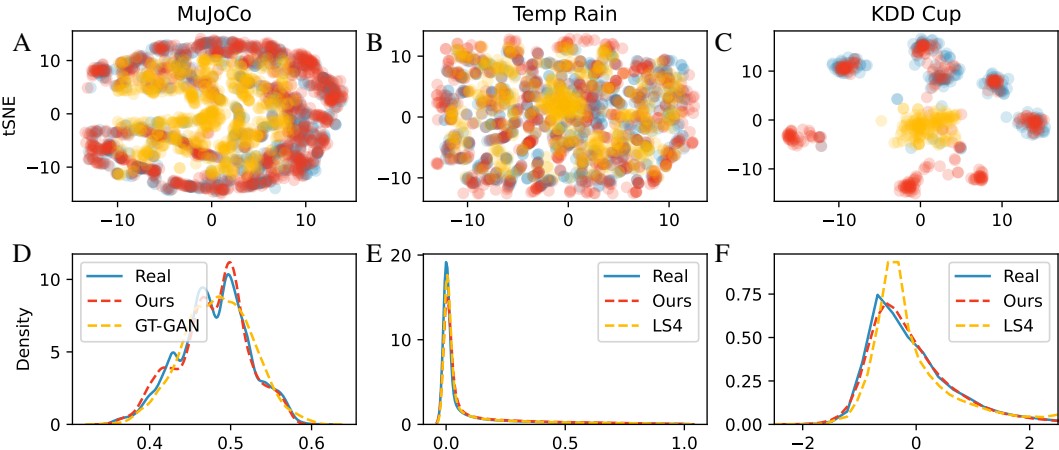

Figure 2: We plot the 2D t-SNE embeddings of synthetic data generated with our method and SOTA tools vs. the real data (top). Then, we compare their probability density functions (bottom).

Table 2: Long time series unconditional marginal, classification, and prediction tasks' results.

| Method | FRED-MD | | | NN5 Daily | | | Temp Rain | | |
|---|---|---|---|---|---|---|---|---|---|
| | marg↓ | class ↑ | pred ↓ | marg↓ | class ↑ | pred ↓ | marg↓ | class ↑ | pred ↓ |
| RNN-VAE | .132 | .036 | 1.47 | .137 | .000 | .967 | .017 | .000 | 159 |
| GP-VAE | .152 | .016 | 2.05 | .117 | .002 | 1.17 | .183 | .000 | 2.31 |
| ODE$^2$VAE | .122 | .028 | .567 | .211 | .001 | 1.19 | 1.83 | .000 | 1.13 |
| Latent ODE | .042 | .327 | .013 | .107 | .000 | 1.04 | **.011** | .000 | 145 |
| TimeGAN | .081 | .029 | .058 | .040 | .001 | 1.34 | .498 | .003 | 1.96 |
| SDEGAN | .084 | .501 | .677 | .085 | .085 | 1.01 | .990 | .017 | 2.46 |
| SaShiMi | .048 | .001 | .232 | .020 | .045 | .849 | .758 | .000 | 2.12 |
| LS4 | .022 | .544 | .037 | .007 | .636 | **.241** | .083 | .976 | .521 |
| Ours | **.021** | **.862** | **.009** | **.005** | **1.02** | .393 | .409 | **5.80** | **.377** |

## 5.2  Long-Term Unconditional Generation

**Data, baselines, and metrics.**   We utilize the long-term time series benchmark presented in [103]. It includes three long-term real-world time series datasets obtained from the Monash Time Series Forecasting Repository [29]: FRED-MD, NN5 Daily, and Temperature Rain. We omit Solar Weekly as it is short-term. These datasets were chosen based on their average 1-lag autocorrelation metric, measuring their correlation over time. Their 1-lag values range from $0.38$ to $0.98$, highlighting a diverse range of temporal dynamics that present challenges for generative learning tasks. Each dataset contains approximately 750 time steps. We compare our method with state-of-the-art long-term generative methods: LS4 [103], SaShiMi [30], SDEGAN [50], TimeGAN [99], Latent ODE [75], ODE$^2$VAE [98], GP-VAE [27] and RNN-VAE [18]. Three different metrics are used to evaluate the generative performance: Marginal (marg), Classification (class), and Prediction (pred). Marginal scores measure the absolute difference between the empirical probability density functions of two distributions. Classification scores use a sequence model to classify samples as real or generated; high scores indicate less distinguishable samples. Prediction scores utilize a train-on-synthetic-test-on-real sequence-to-sequence model to predict future steps; lower scores indicate higher predictability. We used the STFT transform in all our experiments, creating a $32 \times 32$ size image. The number of sampling steps is $18$ and we use the EDM model [45] as the diffusion generative backbone.

**Quantitative and qualitative results.**   We present the results in Tab. 2. LS4 performs well across most datasets and metrics. In comparison, our method outperforms LS4 and the other techniques in almost all cases. On NN5 Daily pred and on Temp Rain marg we achieve inferior results. Notably, Zhou et al. [103] discuss the challenge of measuring the marginal score for the Temp Rain dataset due to frequent zero values. We highlight that our approach substantially improves the classification and prediction scores for the Temp Rain dataset. Additionally, our framework achieves strong results in the classification scores for the FRED-MD and NN5 Daily datasets. We report our results with standard deviations in App. C.4; the results emphasize the statistical significance improvement our framework achieves. Our qualitative results are shown in Fig. 2(B, D) and in App. C.3.

### 5.3 Ultra-long Term Unconditional Generation

**Data, baselines, and metrics.** We conclude our unconditional generation evaluation by considering the challenging setting of *ultra-long* sequences. As far as we know, this setup is underexplored in the literature, and moreover, considering short, long, and ultra-long time series for a single framework is novel to our work. Specifically, we use the following real-world datasets from the Monash Time Series Forecasting Repository [29]: San Francisco Traffic (Traffic) [53] and KDD-Cup 2018 (KDD-Cup) [61]. The datasets' lengths are 17544 and 10920, respectively. Traffic includes an hourly time series detailing the road occupancy rates on the San Francisco Bay Area freeways from 2015 to 2016. KDD-Cup represents the air quality level from 2017 to 2018 estimated by 59 stations across two cities, Beijing (35 stations) and London (24 stations), measured in an hourly rate. We process Traffic with the delay embedding transform ($n = 144, m = 136$), yielding $144 \times 144$ images. KDD-Cup is transformed by STFT, resulting in $112 \times 112$ images. The sampling steps are 36 in both datasets.

**Quantitative and qualitative results.** As shown in Tab. 3, our method consistently achieves superior results in all cases. Notably, it attains on KDD-Cup a pred score of .001 compared to LS4's second-best score of .049. These results highlight our framework's scalability to very long sequences, demonstrating impressive performance across all sequence lengths as we demonstrated in the previous sections. We also report results with standard deviations in

Table 3: Ultra-long unconditional generation.

| Method | Traffic | | | KDD-Cup | | |
|---|---|---|---|---|---|---|
| | pred ↓ | class ↑ | marg ↓ | pred ↓ | class ↑ | marg ↓ |
| Latent-ODE | 1.01 | .000 | .180 | .079 | .013 | .009 |
| LS4 | .170 | .630 | .002 | .049 | .488 | .002 |
| Ours | **.138** | **.684** | **.001** | **.001** | **.842** | **.001** |

App. C.6, emphasizing the statistical significance of our framework. Finally, our qualitative results for this setting are shown in Fig. 2(C, E) and in App. C.3.

### 5.4 Conditional Generation of Time Series

In addition to the unconditional generation benchmark we consider above, we also evaluate our approach on conditional generation tasks. We focus on the imputation (interpolation) and forecasting (extrapolation) tasks, following the experimental setup in [75, 78]. Our approach can be adapted to solve these tasks via a simple modification. For instance, in the interpolation task, the goal is to generate the missing values. Thus, we apply our diffusion model only in the missing locations using a corresponding mask. The rest of the values are left unchanged. A similar mechanism can be applied to extrapolation. Generally, this approach is similar to image inpainting techniques [60]. In the interpolation task, we randomly mask 50% of the sequence values, whereas in the extrapolation challenge, we split the sequence in half, where the second half represents the target values. In this benchmark, we consider short, long, and ultra-long sequences. Our comparison focuses on generative methods that can handle long-range dependencies including ODE-RNN [75], Latent ODE [75], CRU [78], and LS4 [103]. Further details about the experiments can be found in App. B.

**Datasets.** In the short-term setting, we use ETT* datasets [102], that contain electricity loads of various resolutions (ETTh1, ETTh2, and ETTm1, ETTm2) from two electricity stations. The sequence length is 96. For the long-term case, we utilize an established benchmark [75, 78, 103], including the Physionet and USHCN datasets. The Physionet dataset [80] includes health measurements of 41 sensors collected from 8000 ICU patients within the first 48 hours of admission. The United States Historical Climatology Network (USHCN) [63] consists of daily measurements from 1218 weather stations across the United States, including data on precipitation, snowfall, snow depth, and minimum and maximum temperatures. For the ultra-long setting, we use the datasets mentioned in Sec. 5.3.

**Results.** The results of the conditional generation benchmark are detailed in Tab. 4. Values represent the mean squared error (MSE), and thus, lower is better. MSE values are multiplied by $\times 10^{-3}$ and $\times 10^{-2}$ for Physionet and USHCN, respectively, in both experiments. We denote in bold the best method per dataset. The short, long, and ultra-long results are placed at the top, middle, and bottom sections of the table. Overall, our method presents stellar results in all settings, except for ETTm1 where it is second-best. Notably, we mention that in the short interpolation, our results are $\approx 4$ times better than the second-best method, CRU. Similarly, we improve the SOTA by $\approx 30\%$ in the short extrapolation. Our results are particularly strong in the long interpolation setting, where we

Table 4: Interpolation and extrapolation results on datasets of varying lengths. The asterisk (*) denotes non-converging runs, running for over seven days.

| Dataset | Interpolation | | | | | Extrapolation | | | | |
|---|---|---|---|---|---|---|---|---|---|---|
| | ODE-RNN | Latent ODE | CRU | LS4 | Ours | ODE-RNN | Latent ODE | CRU | LS4 | Ours |
| ETTh1 | .210 | .671 | .283 | .642 | **.069** | - | 1.02 | 1.02 | 3.42 | **.701** |
| ETTh2 | .182 | .712 | .368 | 3.40 | **.058** | - | 1.17 | 1.09 | 3.83 | **.667** |
| ETTm1 | .762 | .502 | .086 | .114 | **.038** | - | **.592** | .643 | 3.05 | .634 |
| ETTm2 | .116 | .247 | .179 | .488 | **.049** | - | .414 | .378 | 3.64 | **.348** |
| Physionet | 2.30 | 2.12 | 1.82 | .620 | **.004** | 3.01 | 4.21 | 6.29 | 4.94 | **1.34** |
| USHCN | 8.31 | 17.9 | .160 | .050 | **.006** | 1.96 | 2.03 | 1.27 | 4.36 | **1.20** |
| Traffic | .404 | .985 | * | .990 | **.090** | - | 1.01 | * | 2.23 | **.221** |
| KDD-Cup | .205 | .847 | .190 | .970 | **.144** | - | .696 | .723 | 6.51 | **.368** |

improve by two- and one-orders of magnitude on Physionet and USHCN, respectively. Finally, we also highlight our ultra-long interpolation results which are $\approx 4$ times better than ODE-RNN. We conclude that our approach shows robustness to varying sequence lengths, presenting extremely strong results across several datasets and in comparison to state-of-the-art generative models.

## 5.5 Ablation Studies

We conclude our empirical section by thoroughly inspecting different components of our framework. Specifically, we show that our approach is robust to different image resolutions (App. C.8). We also experiment with a range of hyper-parameters (App. C.9), demonstrating the stability of our approach. Our performance evaluation highlights that our method is comparable to LS4 in terms of training and inference time (App. C.10). Below, we ablate the effect of various image transforms on the performance in the unconditional test. We evaluate our model using four different transforms: folding, Gramian angular field (GAF), delay embedding (DE) and STFT, and we detail the results in Tab. 5. While DE and STFT are slightly better on short and long sequences, respectively, we emphasize that all other transforms perform reasonably well across the various datasets and metrics. GAF does not scale to long sequences as it produces huge images, and thus, it is omitted from the long-term test. We conclude that our framework is robust to the choice of image transformation.

Table 5: Short- and long-term ablation of various image transforms using several datasets and metrics.

| | Energy | | MuJoCo | | FRED-MD | | | NN5 Daily | | |
|---|---|---|---|---|---|---|---|---|---|---|
| | disc↓ | pred↓ | disc↓ | pred↓ | marg↓ | class↑ | pred↓ | marg↓ | class↑ | pred↓ |
| Folding | .074 | **.250** | .017 | **.031** | **.012** | **1.67** | .021 | .010 | .776 | .436 |
| GAF | .349 | .269 | .049 | .034 | - | - | - | - | - | - |
| DE | **.040** | **.250** | **.007** | .033 | .017 | 1.65 | .021 | .007 | **.871** | .394 |
| STFT | .271 | .256 | .071 | .033 | .021 | .862 | **.009** | **.005** | .822 | **.307** |

## 6 Conclusion

While new generative models for general time series data appear rapidly, the majority of existing frameworks are specifically designed to process either short or long sequences. The lack of a unified framework for varying lengths time series can be justified by the shortcomings of current available tools: gradient issues of recurrent networks, temporal computational costs of transformers, and limited expressiveness of state space models. In this work, we address this problem by introducing a novel generative model for time series based on signal-to-image invertible transforms and a vision diffusion backbone. The benefit of our approach is threefold: we exploit advanced diffusion models for vision, we seamlessly process short-to-ultra-long sequences, and we can utilize tools from the signal-to-image literature. We extensively evaluate our framework in the unconditional and conditional settings using short, long, and ultra-long sequences, considering multiple datasets, and in comparison to state-of-the-art models. Our experiments show the superiority of our framework, setting new SOTA results. Further, we demonstrate the robustness of our method through several ablation studies. Our approach requires slightly higher computational resources, which we leave for further consideration and future work. Finally, we believe that the proposed framework has the potential to be applicable in additional tasks including classification, anomaly detection, few-shot learning, and more generally, serve as a foundation model.

## Acknowledgements

This research was partially supported by the Lynn and William Frankel Center of the Computer Science Department, Ben-Gurion University of the Negev, an ISF grant 668/21, an ISF equipment grant, and by the Israeli Council for Higher Education (CHE) via the Data Science Research Center, Ben-Gurion University of the Negev, Israel.

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

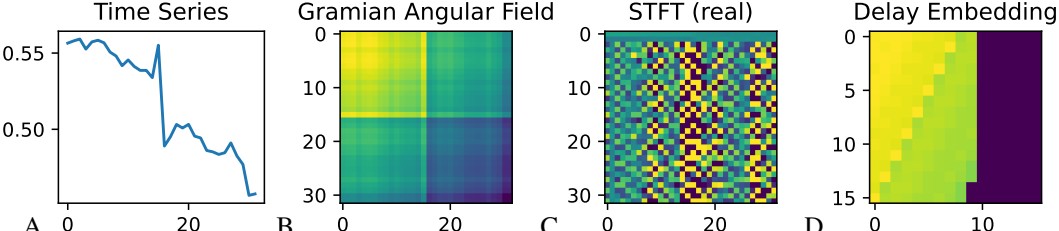

Figure 3: We plot above a time series signal (A), and its image transformations via the Gramian angular field (B), STFT (C), and the delay embedding (D).

## A Domain Transformations

We give a detailed description for each domain transformation and its corresponding forward and inverse processes. We present in Fig. 3 a visual example of a time series and its various different image transformations.

**Folding** is a simple naïve transformation. Given a time series $x$, we fold it into an image $x_{\text{img}}$ by starting from the first row on the left and continuing to the right, jumping to a new row whenever reaching the end of a row. Finally, if needed, we pad with zeroes at the end of the image. The inverse transformation back to the time series is simply taking the in-padded area of the image and unfolding it back to the time series. Although it is simple, this transformation can scale to very long sequences. Folding can be viewed as a particular example of delay embedding, as we detail below.

**Gramian Angular Field** is introduced in [95] for subsequent imputation and classification tasks. It depicts a time series within a polar coordinate system rather than the usual Cartesian coordinates. In the Gramian matrix, each element corresponds to the cosine of the sum of angles. The inverse action, from the image to the time series, is to take the main diagonal. While being a very informative transformation, a major constraint of this transformation is that the height and the width are linear with the size of the time series, preventing it from scaling to long sequences.

**Delay Embedding** [87] transforms a univariate time series $x_{1:L} \in \mathbb{R}^L$ into an image by organizing the series' information into columns and padding as necessary. The hyperparameters for this transformation are $m$ and $n$, where $m$ represents the skip value and $n$ is the column dimension. For a given arbitrary channel of a time series, the transformation constructs the matrix $X$ as follows:

$$X = \begin{bmatrix} x_1 & x_{m+1} & \cdots & x_{L-n} \\ \vdots & \vdots & \cdots & \vdots \\ x_n & x_{n+m+1} & \cdots & x_L \end{bmatrix} \in \mathbb{R}^{n \times q} ,$$

where $q = \lceil (L-n)/m \rceil$. The image $x_{\text{img}}$ is created by padding with zeros to meet the neural network input requirements. The example presented here is for a single channel; scaling to multiple dimensions is straightforward by concatenating each matrix $X$ along another channel. Given an input signal $x \in \mathbb{R}^{L \times K}$, the transformation produces an output $x_{\text{img}} \in \mathbb{R}^{K \times n \times q}$. We pad the image with zeroes right after to create an $x_{\text{img}} \in \mathbb{R}^{K \times n \times n}$.

The original time series $x_{1:L}$ can be reconstructed from $x_{\text{img}}$ by taking each marginal progression from the columns of the matrix $X$. There are various methods to reverse the transformation. For instance, if $m = 1$, $x_{1:L}$ is formed by concatenating the first row and the last column of $x_{\text{img}}$. The delay embedding naturally scales to long sequences; for example, setting $m = n = 256$ allows encoding $65k$ sequences with $256 \times 256$ images.

**Short Time Fourier Transform (STFT)** [35] is a widely used transformation that converts a signal from its original time domain into the frequency domain. The process of computing STFT involves dividing a time-domain signal into shorter, fixed-length segments and then applying the Fourier transform to each segment individually. Given an input signal $x \in \mathbb{R}^{L \times K}$, the STFT produces an output $x_{\text{img}} \in \mathbb{R}^{2K \times H \times W}$. In this output, the number of channels is doubled to store both the

real and imaginary parts of the transformed signal, and $H$ and $W$ are determined by user-defined parameters. These parameters include *n_fft*, which specifies the size of the Fourier transform, and *hop_length*, which defines the distance between successive sliding window frames. Unlike typical audio processing practices, we do not compute the magnitude spectrogram from the STFT output. Instead, we retain both the real and imaginary components within the image, thereby avoiding the need for additional complex spectrogram estimation. This approach maintains the integrity of the full spectral information. After obtaining the STFT images, we normalize them to the range $[-1, 1]$, ensuring that the data is scaled appropriately for subsequent processing.

## B  Experimental Setting

We use the same architectural backbone for all experiments: EDM [45]. We use the AdamW optimizer and train for 1000 epochs, although in practice, all models converged in the range 300-500 epochs. For each task, we elaborate on specific variations in settings and hyperparameters and provide additional information on the training and evaluation protocol.

### B.1  Short-term unconditional generation.

**Data.**   For the short-term unconditional generation (Sec. 5.1), we utilize four synthetic and real-world datasets with a fixed length of $24$: *Stocks*, consisting of daily historical Google stock data from 2004 to 2019, comprising six channels: high, low, opening, closing, and adjusted closing prices, as well as volume. This data lacks periodicity and is dominated by random walks. The second dataset, *Energy*, is a multivariate appliance energy prediction dataset [14], featuring 28 channels with correlated features, and it exhibits noisy periodicity and continuous-valued measurements. The third dataset, *MuJoCo* (Multi-Joint dynamics with Contact), serves as a versatile physics generator for simulating TS data with 14 channels [89]. The last dataset, *Sine*, is a multivariate simulated dataset, where each sample $x_t^i(j)$ is defined as $\sin(2\pi\eta t + \theta)$, where $\eta$ is sampled from a uniform distribution $[0, 1]$ and $\theta$ is sampled from a uniform distribution $[-\pi, \pi]$, with five channels for $j$.

**Hyperparameters.**   We describe below in Tab. 6 the different hyperparameters used in our framework. For all datasets, we used the same default sampler of EDM [45], and we mention the hyperparameters of their U-net model that we tune in our work. Please see [45] for further details about the U-net model hyperparameters.

Table 6: Short-term unconditional generation hyperparameters including short time Fourier transform (STFT), delay embedding (DE) hyperparameters and diffusion hyperparameters

|  | **Stocks** | **Energy** | **MuJoCo** | **Sine** |
| --- | --- | --- | --- | --- |
| **General** | | | | |
| *image size* | $8 \times 8$ | $8 \times 8$ | $8 \times 8$ | $8 \times 8$ |
| *learning rate* | $10^{-4}$ | $10^{-4}$ | $10^{-4}$ | $10^{-4}$ |
| *batch size* | 128 | 128 | 128 | 128 |
| **DE** | | | | |
| *embedding (n)* | 8 | 8 | 8 | 8 |
| *delay (m)* | 3 | 3 | 3 | 3 |
| **STFT** | | | | |
| *n_fft* | - | - | - | - |
| *hop_length* | - | - | - | - |
| **Diffusion** | | | | |
| *U-net channels* | 128 | 128 | 64 | 128 |
| *in channels* | $[1, 2, 2, 2]$ | $[1, 2, 2, 4]$ | $[1, 2, 2, 2]$ | $[1, 2, 2, 2]$ |
| *sampling steps* | 18 | 18 | 18 | 18 |

**Evaluation.**   We utilize the benchmark proposed in [99] to evaluate short-term unconditional generation and adhere to its evaluation protocol. This protocol comprises two scores: a predictive score and a discriminative score. The predictive score assesses the utility of the generated data by

training an independent prediction model on the generated data; superior generations result in better prediction scores for this model. The discriminative score evaluates the similarity of distributions using a proxy discriminator trained to distinguish between generated and original samples; higher scores indicate that the generative model has accurately captured the underlying distribution of the data. For more details about the evaluation protocol, please refer to [19] or [99].

## B.2 Long-term unconditional generation.

**Data.** In our exploration of long-term unconditional generation, we employ the benchmark for long-term time series data as presented in [103]. This benchmark encompasses three extensive real-world time series datasets from the Monash Time Series Forecasting Repository [29]: FRED-MD, NN5 Daily, and Temperature Rain. These datasets were meticulously selected based on their average 1-lag autocorrelation metric, which quantifies the 1-step correlation over time. The 1-lag values, ranging from 0.38 to 0.98, exemplify a broad spectrum of temporal dynamics, thereby presenting significant challenges for generative learning models. To ensure uniformity in the NN5 Daily and FRED-MD datasets, each sequence within these datasets is normalized such that each trajectory is centered at its mean and adheres to a normal distribution. This normalization approach is advantageous for datasets like NN5 Daily, where the minimum and maximum values can vary substantially across different data points. For the Temperature Rain dataset, sequences are scaled to the [0, 1] range, considering the data's consistently positive values and its tendency to cluster around the x-axis with occasional sharp spikes. Each dataset comprises approximately 750 time steps, providing a robust basis for evaluating long-term generative performance.

**Hyperparameters.** We describe below in Tab. 7 the different hyperparameters used in our framework. For all datasets, we used the same default sampler of EDM [45], and we mention the hyperparameters of the U-net model that we tune in our work, please see [45] for more details about these hyperparameters. In addition, For all long-term experiments, we use the AdamW optimizer with a weight decay of $10^{-5}$.

Table 7: Long-term unconditional generation hyperparameters including short time Fourier transform (STFT), delay embedding (DE) hyperparameters and diffusion hyperparameters

|  | **Fred-MD** | **Temperature Rain** | **NN5 Daily** |
|---|---|---|---|
| **General** | | | |
| *image size* | $32 \times 32$ | $32 \times 32$ | $32 \times 32$ |
| *learning rate* | $10^{-4}$ | $10^{-4}$ | $10^{-4}$ |
| *batch size* | 32 | 64 | 32 |
| **DE** | | | |
| *embedding(n)* | — | — | — |
| *delay(m)* | — | — | — |
| **STFT** | | | |
| *n_fft* | 63 | 63 | 63 |
| *hop_length* | 23 | 23 | 25 |
| **Diffusion** | | | |
| *U-net channels* | 128 | 128 | 128 |
| *in channels* | $[1, 2, 4, 4]$ | $[1, 2, 4, 4]$ | $[1, 2, 4, 4]$ |
| *sampling steps* | 18 | 18 | 18 |

**Evaluation.** To assess model performance, we follow the benchmark used in [103]. Our evaluation comprises classification and prediction models, each employing linear encoders and decoders with a single S4 layer having 16 hidden state dimensions. In the classification model, the encoder maps data dimensions to 16 hidden states. The S4 layer's output sequence is averaged before being passed to the decoder, which produces logits for binary classification using cross-entropy loss. Similarly, the prediction model's encoder maps input to a 16-dimensional hidden state, while the decoder maps it back to the original data dimension, predicting $k = 10$ future steps. Both models are trained using the AdamW optimizer, which has a learning rate of 0.01 over 100 epochs and a batch size of 128. The optimizer generates samples equal to testing data points to train the models together.

### B.3 Ultra-long-term unconditional generation.

**Data.**    For the ultra-long-term unconditional task, we introduce a novel benchmark consisting of two datasets: San Francisco Traffic (Traffic) [53] and KDD-Cup 2018 (KDD-Cup) [61]. The datasets' lengths are 17544 and 10920, respectively. Traffic includes an hourly time series detailing the road occupancy rates on the San Francisco Bay Area freeways from 2015 to 2016. KDD-Cup represents the air quality level from 2017 to 2018 estimated by 59 stations across two cities, Beijing (35 stations) and London (24 stations), measured in an hourly rate. We follow the same normalization procedure applied to Fred-MD and NN5 daily, as described in B.2.

**Hyperparameters.**    In Tab. 8 below, we outline the various hyperparameters used in our framework. For all datasets, we employed the default sampler of EDM [45], and we specified the U-net model hyperparameters that we tuned in our study. For more information on the U-net model hyperparameters, please refer to [45].

Table 8: Ultra-long-term unconditional generation hyperparameters including short time Fourier transform (STFT), delay embedding (DE) hyperparameters and diffusion hyperparameters

|  | **Traffic** | **KDD-Cup** |
|---|---|---|
| **General** | | |
| *image size* | $144 \times 144$ | $112 \times 112$ |
| *learning rate* | $10^{-4}$ | $10^{-4}$ |
| *batch size* | 8 | 16 |
| **DE** | | |
| *embedding(n)* | 144 | – |
| *delay(m)* | 136 | – |
| **STFT** | | |
| *n_fft* | – | 223 |
| *hop_length* | – | 98 |
| **Diffusion** | | |
| *U-net channels* | 128 | 128 |
| *in channels* | $[1, 2, 4, 4]$ | $[1, 2, 4, 4]$ |
| *sampling steps* | 18 | 18 |

**Evaluation.**    For the ultra-long-term unconditional generation task, we follow the same procedure outlined in B.2. We use the same classification and prediction models, as they effectively distinguish between low- and high-quality ultra-long-term generations.

### B.4 Conditional generation

**Data.**    For short-term interpolation and extrapolation benchmarks (Sec. 5.4), we use the *ETT\** datasets [102], each with a fixed length of 96. The ETT datasets are crucial indicators for long-term electric power deployment, containing two years of data from two separate counties in China. The datasets are divided into *ETTh1* and *ETTh2* for 1-hour intervals, and *ETTm1* and *ETTm2* for 15-minute intervals. Each data point includes the target value "oil temperature" and six power load features.

For long-term interpolation and extrapolation, we employ a well-established benchmark [75, 78, 103], incorporating the Physionet and USHCN datasets. The data extraction for the USHCN dataset follows the procedure detailed by [20]. Notably, both datasets exhibit sparsity across many features and contain numerous zero values. The Physionet dataset [80] includes health measurements from 41 sensors collected from 8,000 ICU patients within the first 48 hours of admission. The United States Historical Climatology Network (USHCN) [63] provides daily measurements from 1,218 weather stations across the United States, covering precipitation, snowfall, snow depth, and minimum and maximum temperatures. For the conditional tasks, we strictly follow the training and evaluation procedures described by [78], and we refer readers to this work for a comprehensive explanation of the evaluation protocol. Both datasets span approximately 1,000 to 2,000 time steps.

For the ultra-long-term conditional generation task, we utilize the same data used in the ultra-long-term unconditional generation benchmark, and we refer to App. B.3 for more details about the datasets.

**Hyperparameters.** We describe below in Tab. 9 the different hyperparameters used in our framework. The hyperparameters are similar for both tasks. Therefore, we present them in a unified table. For all datasets, we used the same default sampler of EDM[45]; we also present the hyperparameters of the U-net model; for further details about them, please see [45].

Table 9: Conditional generation hyperparameters including short time Fourier transform (STFT), delay embedding (DE) hyperparameters and diffusion hyperparameters

| | ETTh1 | ETTh2 | ETTm1 | ETTm2 | Physionet | USHCN | Traffic | KDD-Cup |
|---|---|---|---|---|---|---|---|---|
| **General** | | | | | | | | |
| *image size* | $32 \times 32$ | $32 \times 32$ | $32 \times 32$ | $32 \times 32$ | $32 \times 32$ | $32 \times 32$ | 144 | 128 |
| *learning rate* | $10^{-4}$ | $10^{-4}$ | $10^{-5}$ | $30^{-5}$ | $10^{-5}$ | $10^{-4}$ | $10^{-4}$ | $10^{-4}$ |
| *batch size* | 32 | 32 | 16 | 64 | 8 | 8 | 8 | 8 |
| **DE** | | | | | | | | |
| *embedding(n)* | 32 | 32 | 32 | 32 | 32 | 32 | 144 | 128 |
| *delay(m)* | 3 | 3 | 3 | 3 | 30 | 30 | 122 | 86 |
| **STFT** | | | | | | | | |
| *n_fft* | – | – | – | – | – | – | – | – |
| *hop_length* | – | – | – | – | – | – | – | – |
| **Diffusion** | | | | | | | | |
| *U-net channels* | 128 | 128 | 64 | 128 | 128 | 128 | 128 | 128 |
| *in channels* | $[1,2,2,2]$ | $[1,2,2,4]$ | $[1,2,4,8]$ | $[1,2,4,8]$ | $[1,2,4,4]$ | $[1,2,4,4]$ | $[1,2,4,4]$ | $[1,2,4,4]$ |
| *sampling steps* | 18 | 18 | 18 | 18 | 18 | 18 | 36 | 36 |

**Evaluation.** For the short-term and ultra-long-term datasets (ETT*, Traffic, KDD-Cup), we follow the next procedure. In the interpolation task, we randomly mask $50\%$ of the input data and train the models to predict the missing masked $50\%$. In the extrapolation task, we mask the second half of the sequence and train the models to predict this missing half. We measure the distance between the models' generated outcomes and the ground truth using MSE loss. Accurate generation will lead to a smaller distance between the prediction and the ground truth, thus indicating the models' interpolation and extrapolation capabilities. For the long-term datasets, we follow a similar procedure as above; however, since the data is sparse and irregularly sampled, the masking is slightly different. We adhere to the exact interpolation and extrapolation processes described in [78] and refer to that source for more details.

## C    Additional Experiments and Analysis

### C.1    Short-term unconditional generation

Due to space constraints in the main paper, we report the rest of the benchmark here. In Tab. 10, we show our model performance on the simple toy **Sine** dataset.

### C.2    Wasserstein distance analysis

In Tab. 11, we present the Wasserstein distances calculated between our generated 2D point cloud and the actual data. A lower score indicates greater similarity between the clusters, meaning that a lower score is preferable. Our approach yields the best scores across all datasets in comparison to GT-GAN on short sequences and LS4 on long and ultra-long time series.

### C.3    Short-term unconditional generation qualitative analysis

We include the *Stocks*, *Energy* and the *MuJoCo* qualitative t-SNE evaluation (Fig. 4(A,B,C)) and density analysis (Fig .4(D, E, F)). In addition, we show in Tab. 11 a quantitative evaluation of the t-SNE clusters Wasserstein distance. Both the visual results and the quantitative results demonstrate our framework's ability to learn the true distribution across multiple datasets.

Table 10: Short time series unconditional generation task on the Sine dataset.

| Method | Sine disc↓ | Sine pred↓ |
|---|---|---|
| KoVAE | **.005 ± .003** | **.093 ± .000** |
| DiffTime | .013 ± .006 | **.093 ± .000** |
| GT-GAN | .012 ± .014 | .097 ± .000 |
| TimeGAN | .011 ± .008 | .093 ± .019 |
| RCCGAN | .022 ± .0068 | .097 ± .001 |
| C-RNN-GAN | .229 ± .040 | .127 ± .004 |
| WaveNet | .158 ± .011 | .117 ± .008 |
| WaveGAN | .277 ± .013 | .134 ± .013 |
| LS4 | .342 ± .007 | .132 ± .011 |
| Ours | .014 ± .009 | .094 ± .000 |

Table 11: We calculate the Wasserstein distances between the original cluster, our generated samples cluster the other method cluster shown in Figs. 4, 5 and 6.

| Method | Stocks | Energy | MuJoCo | Temp Rain | NN5 Daily | Traffic | KDD-Cup |
|---|---|---|---|---|---|---|---|
| GT-GAN | 6.20 | 3.35 | 3.83 | - | - | - | - |
| LS4 | - | - | - | 3.27 | 5.23 | 4.63 | 11.84 |
| Ours | **2.84** | **3.20** | **1.05** | **2.85** | **5.21** | **3.19** | **6.59** |

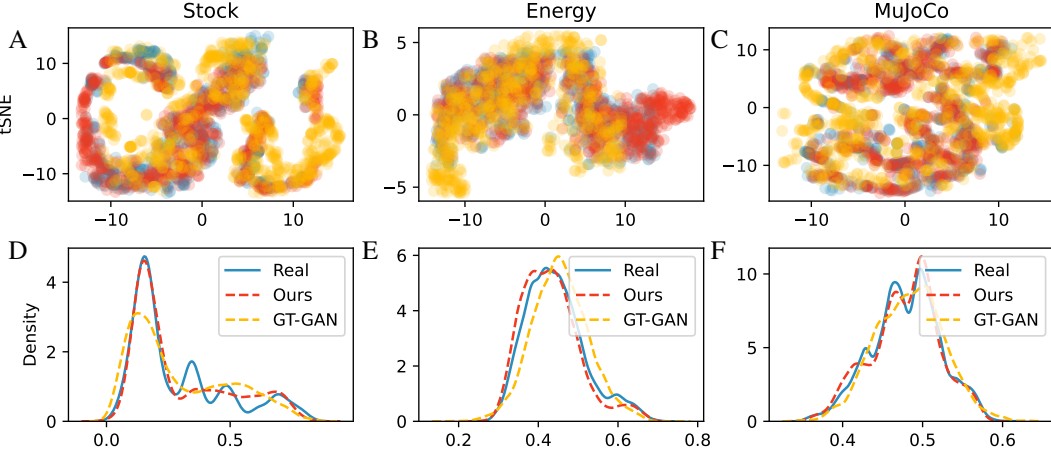

Figure 4: We plot the 2D t-SNE embeddings of synthetic data generated with our method and SOTA tools vs. the real data (top). Then, we compare their probability density functions (bottom).

## C.4 Long-term unconditional generation with standard deviation

In Tab. 12, we present the results of our method on unconditional generation of long sequences with standard deviation. The results demonstrate our method's statistical significance compared to the state-of-the-art method LS4.

Table 12: Long time series unconditional generation task with standard deviation.

| | LS4 marg↓ | LS4 class ↑ | LS4 pred ↓ | Ours marg↓ | Ours class ↑ | Ours pred ↓ |
|---|---|---|---|---|---|---|
| FRED-MD | .022 | .544 | .037 | **.021 ± .000** | **.862 ± .227** | **.009 ± .003** |
| NN5 Daily | .007 | .636 | **.241** | **.005 ± .000** | **.822 ± .157** | .307 ± .037 |
| Temp Rain | **.083** | .976 | .521 | .409 ± .000 | **5.80 ± .974** | **.377 ± .022** |

## C.5 Long-term unconditional generation qualitative analysis

We include the qualitative t-SNE evaluation for *Temp Rain* and *NN5 Daily* (Fig. 5(A, B)) and their density analysis (Fig. 5(D, E)). Additionally, we provide in Tab. 11 the quantitative evaluation of the t-SNE clusters using the Wasserstein distance. Our results indicate the superiority of our approach in comparison to other techqniques.

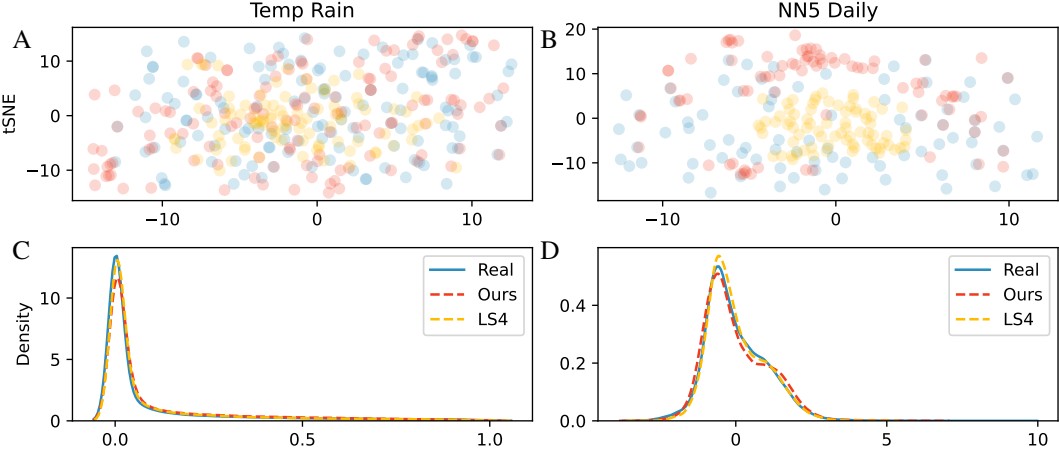

Figure 5: We plot the 2D t-SNE embeddings of synthetic data generated with our method and SOTA tools vs. the real data (top). Then, we compare their probability density functions (bottom).

## C.6 Ultra-long time series unconditional generation with standard deviation

In Tab. 13, we present the results of our method for unconditional generation of ultra-long sequences, including standard deviations. These results demonstrate the statistical significance of our method compared to the state-of-the-art competitive methods.

Table 13: Ultra-long unconditional generation with standard devation

| Method | Traffic | | | KDD-Cup | | |
|---|---|---|---|---|---|---|
| | pred $\downarrow$ | class $\uparrow$ | marg $\downarrow$ | pred $\downarrow$ | class $\uparrow$ | marg $\downarrow$ |
| Latent ODE | $1.01 \pm .412$ | $.000 \pm .000$ | $.180 \pm .000$ | $.079 \pm .055$ | $.013 \pm .020$ | $.009 \pm .000$ |
| LS4 | $.170 \pm .030$ | $.630 \pm .060$ | $.002 \pm .000$ | $.049 \pm .046$ | $.488 \pm .164$ | $.002 \pm .000$ |
| Ours | $\mathbf{.138 \pm .014}$ | $\mathbf{.684 \pm .019}$ | $\mathbf{.001 \pm .000}$ | $\mathbf{.001 \pm .000}$ | $\mathbf{.842 \pm .245}$ | $\mathbf{.001 \pm .000}$ |

## C.7 Ultra-long-term unconditional generation qualitative analysis

We include the qualitative t-SNE evaluation for *Traffic* and *KDD-Cup* (Fig. 6(A, B)) and the density analysis (Fig. 6(D, E)). Additionally, we provide in Tab. 11 the quantitative evaluation of the t-SNE clusters using the Wasserstein distance. Our results highlight our method ability to handle very long sequences.

## C.8 Image size ablation

Given the significant impact of image size on the computational resources required by our method, we investigate whether varying image size influences different transformations. We explore the effect of different image sizes on our framework, utilizing long-term datasets with short time Fourier transform (STFT) and short-term datasets with delay embedding. For short time series consisting of 24 time steps, we test image sizes of 8 and 16, as we observe that scaling to larger sizes may not yield benefits. For long time series of approximately 750 steps, we experiment with sizes of 32, 64, and 128. Notably, 32 is the minimum size required to contain enough pixels for representing the long sequence adequately. We present the results in Tab. 14. While most results demonstrate high

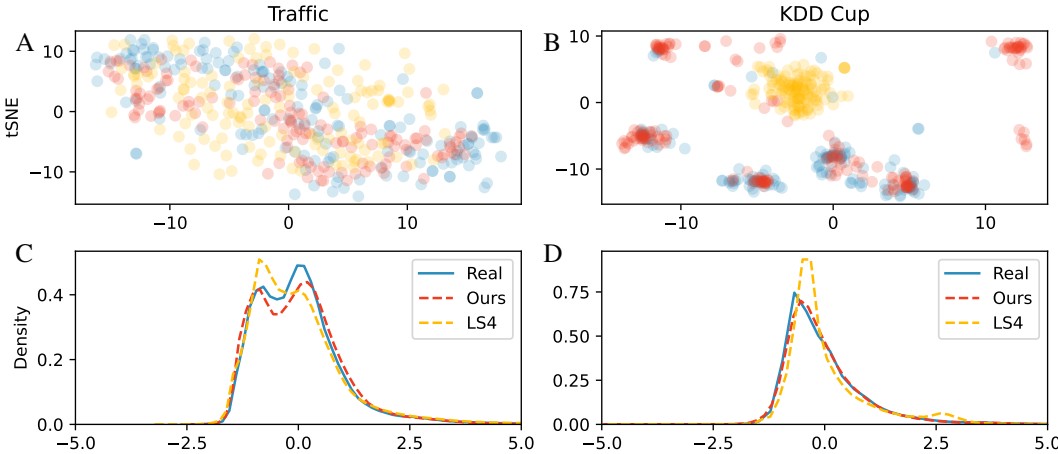

Figure 6: We plot the 2D t-SNE embeddings of synthetic data generated with our method and SOTA tools vs. the real data (top). Then, we compare their probability density functions (bottom).

competitiveness compared to other methods, inconclusive experimental results indicate that image size does not significantly affect generation quality. Therefore, it is reasonable to use the minimum length that a transformation can accommodate to benefit from minimum computational costs.

Table 14: Image size ablation study.

| Image Size | FRED-MD | | | NN5 Daily | | | Energy | | MuJoCo | |
|---|---|---|---|---|---|---|---|---|---|---|
| | marg↓ | class ↑ | pred ↓ | marg↓ | class ↑ | pred ↓ | disc↓ | pred ↓ | disc↓ | pred ↓ |
| Short Series | | | | | | | | | | |
| 8 x 8 | - | - | - | - | - | - | **.040** | **.250** | **.007** | .033 |
| 16 x 16 | - | - | - | - | - | - | .059 | **.250** | .036 | **.032** |
| Long Series | | | | | | | | | | |
| 32 x 32 | .021 | .862 | **.009** | **.005** | .822 | **.307** | - | - | - | - |
| 64 x 64 | .021 | **1.82** | .021 | .008 | **.829** | .440 | - | - | - | - |
| 128 x 128 | **.016** | 1.65 | .023 | .012 | .733 | .430 | - | - | - | - |

### C.9 Hyeprparameters Ablation

**Ablation study on diffusion sampling steps.** While [45] demonstrate an improvement in FID score with a larger number of steps in their work, we do not observe the same trend in our framework. The results are presented in Tab. 15 in the first section, indicating an unclear trend across different datasets and metrics.

**Ablation study on batch size.** The results in Tab. 15 in the middle section demonstrate that our framework is unaffected by different batch sizes. This is a positive indication of our framework's adaptability to various computational environments, whether with low memory or high memory capabilities.

**Ablation study on learning rate.** In our examination of the learning rate, we have made an intriguing observation. We have found that when the learning rate equals or exceeds $10^{-3}$, the diffusion backbone [45] tends to collapse, resulting in the generation of irrelevant signals. This phenomenon is clearly demonstrated in Tab. 15 in the middle section. With a learning rate of $10^{-3}$, the *disc* scores are .256 and $0.499$ for the MuJoCo and Energy datasets, respectively, indicating random generation. However, when the learning rate is lowered, we have not observed any such collapse of the backbone on any dataset or task.

Table 15: Hyperparameters ablation study. We study the effect of different hyperparameters on our framework. We utilize four short and long-term datasets.

| Hyperparameter | FRED-MD | | | NN5 Daily | | | Energy | | MuJoCo | |
|---|---|---|---|---|---|---|---|---|---|---|
| | marg↓ | class ↑ | pred ↓ | marg↓ | class ↑ | pred ↓ | disc↓ | pred ↑ | disc↓ | pred ↑ |
| Diffusion Sampling Steps | | | | | | | | | | |
| 18 | .021 | .862 | .009 | .005 | .822 | .307 | .040 | .250 | .007 | .033 |
| 36 | .015 | 1.33 | .020 | .009 | .829 | .399 | .052 | .250 | .017 | .032 |
| 72 | .017 | 1.36 | .022 | .009 | .836 | .395 | .057 | .250 | .025 | .031 |
| 144 | .018 | 1.32 | .024 | .010 | .836 | .397 | .058 | .250 | .025 | .030 |
| Batch Size | | | | | | | | | | |
| 16 | .022 | 1.41 | .023 | .009 | .804 | .403 | .060 | .250 | .012 | .032 |
| 32 | .018 | 1.31 | .021 | .010 | .850 | .396 | .059 | .250 | .009 | .032 |
| 64 | .019 | 1.32 | .021 | .009 | .842 | .394 | .050 | .250 | .019 | .032 |
| Learning Rate | | | | | | | | | | |
| $10^{-3}$ | .019 | 1.02 | .025 | .012 | .827 | .415 | .499 | .252 | .256 | .043 |
| $10^{-4}$ | .021 | 1.54 | .024 | .010 | .813 | .401 | .065 | .249 | .007 | .033 |
| $10^{-5}$ | .021 | 1.16 | .025 | .007 | 1.01 | .421 | .056 | .250 | .020 | .031 |

## C.10 Computational Resources Comparison

In this section, we compare the computational resources required by our proposed method and the LS4 method, focusing on training and inference wall-clock runtime and model size in terms of parameters, and we analyze the FLOPs used per method. Although our method, which utilizes image transforms and diffusion models, has a larger model size in terms of parameters, it remains comparable to LS4 regarding training and inference time. Despite the larger model size, our method achieves similar training and inference efficiency, making it a viable and scalable solution for large-scale time series generation tasks as shown in Tab.16. Furthermore, the rapid advancements and growing research interest in faster sampling techniques for diffusion models [82] present an opportunity to further enhance our method's efficiency. Leveraging these developments, our approach can integrate even more optimized diffusion models, potentially reducing the computational time and resources required for training and inference, thus improving scalability for large-scale time series generation tasks. Finally, we analyze the FLOPs used in our method compared to LS4 and DiffTime on the Stock, nn5daily and KDD Cup datasets in Tab. 17.

Table 16: Computational resources in terms of training wall-clock runtime (WCR) in minutes(m) or hours(h), and model parameters (MP) in millions (M)

| Method | Stocks | | Energy | | NN5 Daily | | Temp Rain | |
|---|---|---|---|---|---|---|---|---|
| | WCR | MP | WCR | MP | WCR | MP | WCR | MP |
| TimeGAN | 2h 59m | 48K | 3h 37m | 1M | - | - | - | - |
| GT-GAN | 12h 20m | 41K | 10h 39m | 57k | - | - | - | - |
| DiffTime | 52m | 240k | - | - | 46m | 32M | - | - |
| LS4 | 5h 30m | 2.7M | 2h | 2.1M | 53m | 2.1M | 27h | 2.3M |
| Ours | 1h 10m | 575K | 1h | 2M | 58m | 5.9M | 30h | 6.4M |

## C.11 Scaling Laws Analysis

In this section, we investigate how the performance of our proposed method scales with the size of the underlying image diffusion model. Specifically, we evaluate the impact of increasing the model size from a few thousand parameters to several hundred million on various time-series datasets. Additionally, we compare this trend with other state-of-the-art time-series generative models, analyzing how their performance is affected by model size increments. Interestingly, merely increasing the model parameters does not improve their performance. It demonstrates that simply enlarging previous methods does not necessarily enhance their generation capabilities. Moreover, in the case of LS4 on the KDD Cup dataset, increasing the model's parameters to 100 million results

Table 17: FLOPs analysis on DiffTime, LS4 and Our method

| #Params | DiffTime | | | LS4 | | | Ours | | |
|---|---|---|---|---|---|---|---|---|---|
| | Stocks | nn5daily | KDD Cup | Stocks | nn5daily | KDD Cup | Stocks | nn5daily | KDD Cup |
| 500k | 0.057G | 0.751G | 14.485G | 0.003G | 0.107G | 1.480G | 0.009G | 0.123G | 1.480G |
| 1M | 0.037G | 1.301G | 24.394G | 0.007G | 0.233G | 3.225G | 0.014G | 0.217G | 2.624G |
| 5M | 0.747G | 4.289G | 99.402G | 0.048G | 1.594G | 22.015G | 0.070G | 1.084G | 13.21G |
| 25M | 3.838G | 20.864G | 414.15G | 0.191G | 6.299G | 86.966G | 0.291G | 4.683G | 57.25G |
| 50M | 7.996G | 41.060G | 711.04G | 0.352G | 11.602G | 160.172G | 0.633G | 10.06G | 123.0G |
| 100M | 14.544G | 80.198G | 1387.3G | 0.759G | 25.040G | − | 1.335G | 21.27G | 260.3G |
| 150M | 22.407G | 118.865G | 1874.5G | 1.084G | 35.739G | − | 1.890G | 31.58G | 386.6G |

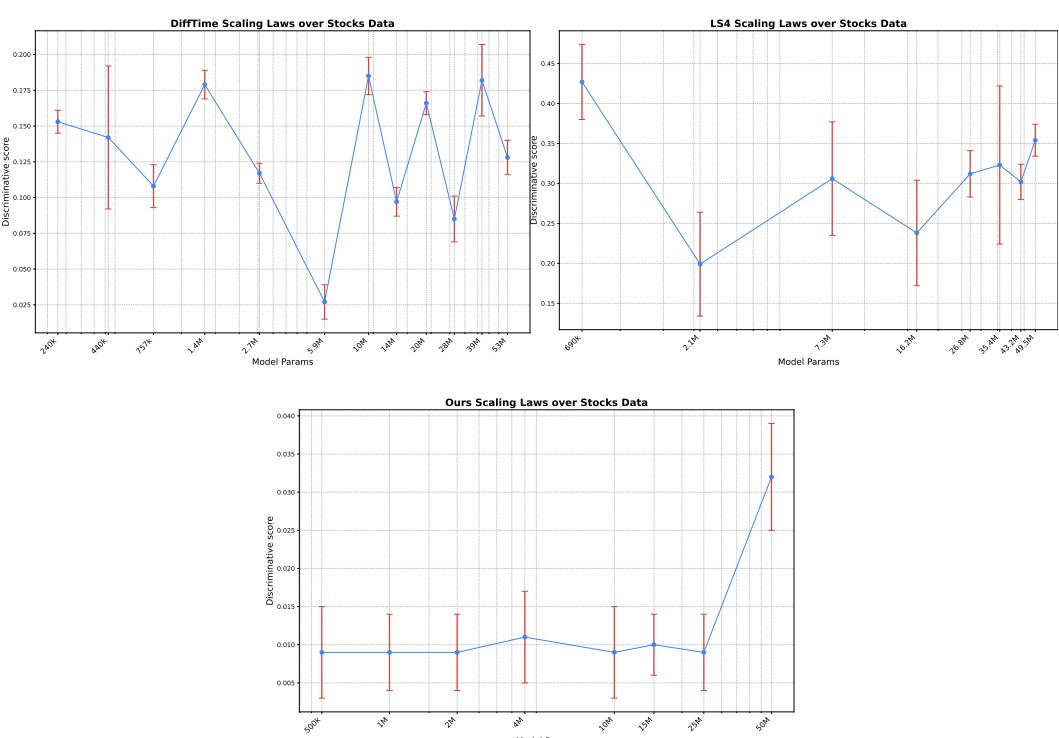

Figure 7: Scaling analysis of different models on Stocks data. A lower discriminative score is better.

in memory collapse, making it infeasible to run a batch size of one with the current resources used for training all models. We present the results for our method in Tab. 18, for LS4 in Tab. 19 and the results for DiffTime, in Tab. 20 and Tab. 21. Finally, we add a visualization of the results for the Stocks dataset in Fig. 7.

Table 18: Our framework scaling laws

| #P | Stocks | | Energy | | nn5daily | | |
|---|---|---|---|---|---|---|---|
| | disc↓ | pred↓ | disc↓ | pred↓ | marg↓ | class↑ | pred↓ |
| 0.5M | $.009 \pm .006$ | $.036 \pm .000$ | $.100 \pm .009$ | $.253 \pm .000$ | .012 | $.765 \pm .275$ | $.407 \pm .640$ |
| 1M | $.009 \pm .005$ | $.036 \pm .000$ | $.087 \pm .005$ | $.251 \pm .000$ | .010 | $1.10 \pm .315$ | $.421 \pm .076$ |
| 2M | $.009 \pm .005$ | $.036 \pm .000$ | $.04 \pm .006$ | $.250 \pm .000$ | .006 | $1.03 \pm .263$ | $.389 \pm .056$ |
| 4M | $.011 \pm .006$ | $.036 \pm .000$ | $.053 \pm .004$ | $.250 \pm .000$ | .005 | $.859 \pm .368$ | $.443 \pm .117$ |
| 10M | $.009 \pm .006$ | $.036 \pm .000$ | - | - | .006 | $.859 \pm .153$ | $.428 \pm .032$ |
| 15M | $.010 \pm .004$ | $.036 \pm .000$ | - | - | - | - | - |
| 25M | $.009 \pm .005$ | $.036 \pm .000$ | - | - | - | - | - |
| 50M | $.032 \pm .007$ | $.036 \pm .000$ | - | - | .005 | $.420 \pm 0.021$ | $.884 \pm 0.223$ |

Table 19: LS4 scaling laws

| #P | Energy | | Stocks | | nn5daily | | |
|---|---|---|---|---|---|---|---|
| | disc↓ | pred↓ | disc↓ | pred↓ | marg↓ | class↑ | pred↓ |
| 0.69M | $.498 \pm .000$ | $.374 \pm .019$ | $.427 \pm .047$ | $.047 \pm .0045$ | .061 | $.002 \pm .003$ | $1.97 \pm 1.17$ |
| 2.1M | $.474 \pm .003$ | $.251 \pm .000$ | $.199 \pm .065$ | $.068 \pm .013$ | .011 | $.719 \pm .138$ | $.305 \pm .054$ |
| 7.3M | $.488 \pm .002$ | $.311 \pm .003$ | $.306 \pm .071$ | $.058 \pm .001$ | .067 | $.005 \pm .009$ | $1.33 \pm .288$ |
| 16.2M | $.489 \pm .003$ | $.262 \pm .001$ | $.238 \pm .066$ | $.038 \pm .000$ | .011 | $.733 \pm .344$ | $.335 \pm .105$ |
| 26.8M | $.494 \pm .002$ | $.311 \pm .003$ | $.312 \pm .029$ | $.054 \pm .001$ | .019 | $.213 \pm .167$ | $.275 \pm .034$ |
| 35.4M | – | – | $.323 \pm .099$ | $.071 \pm .003$ | .008 | $.771 \pm .225$ | $.294 \pm .069$ |
| 43.2M | – | – | $.302 \pm .022$ | $.067 \pm .000$ | – | – | – |
| 49.5M | $.497 \pm .001$ | $.294 \pm .002$ | $.354 \pm .020$ | $.037 \pm .000$ | – | – | – |
| 102.4M | $.497 \pm .001$ | $.276 \pm .001$ | – | – | .090 | $.006 \pm .000$ | $7.07 \pm 2.21$ |
| 152.7M | $.498 \pm .000$ | $.296 \pm .005$ | – | – | .165 | $.001 \pm .000$ | $2.02 \pm .897$ |

Table 20: DiffTime scaling laws

| #P | Stocks | | nn5 daily | | |
|---|---|---|---|---|---|
| | disc↓ | pred↓ | marg↓ | class↑ | pred↓ |
| 240k | $.153 \pm .008$ | $.037 \pm .037$ | .030 | $.179 \pm .131$ | $4.33 \pm 1.54$ |
| 440k | $.142 \pm .050$ | $.038 \pm .038$ | .026 | $.284 \pm .252$ | $2.79 \pm .591$ |
| 757k | $.108 \pm .015$ | $.038 \pm .038$ | .015 | $.421 \pm .316$ | $1.92 \pm .941$ |
| 1.4M | $.179 \pm .010$ | $.037 \pm .037$ | .021 | $.525 \pm .410$ | $11.43 \pm 19.03$ |
| 2.7M | $.117 \pm .007$ | $.039 \pm .039$ | .016 | $.279 \pm .217$ | $.981 \pm .207$ |
| 5.9M | $.027 \pm .012$ | $.037 \pm .037$ | .006 | $.324 \pm .160$ | $.666 \pm .199$ |
| 10M | $.185 \pm .013$ | $.037 \pm .037$ | .016 | $.272 \pm .135$ | $.939 \pm .224$ |
| 14M | $.097 \pm .010$ | $.037 \pm .037$ | .011 | $.308 \pm .199$ | $10.53 \pm 7.06$ |
| 20M | $.166 \pm .008$ | $.037 \pm .037$ | .017 | $.312 \pm .147$ | $.441 \pm .084$ |
| 28M | $.085 \pm .016$ | $.039 \pm .039$ | .019 | $.134 \pm .097$ | $.583 \pm .086$ |
| 39M | $.182 \pm .025$ | $.042 \pm .042$ | .021 | $.182 \pm .155$ | $.681 \pm .187$ |
| 53M | $.128 \pm .012$ | $.037 \pm .037$ | .008 | $.140 \pm .160$ | $.896 \pm .598$ |
| 103M | – | – | .015 | $.345 \pm .132$ | $.584 \pm .211$ |
| 156M | – | – | .018 | $.675 \pm .744$ | $1.700 \pm 1.161$ |

## C.12 Other Image Generative Models

Our goal in this paper is to leverage recent advancements in computer vision to develop an elegant and robust solution for time-series data, addressing different sequence lengths and setting a baseline for handling short, long, and ultra-long sequences. We aim to take advantage of the fact that image architectures are more thoroughly explored. We hypothesize that the improvements we observe are largely due to the more advanced development of image architectures compared to time-series architectures. To further investigate this, we used NVAE [90], and StyleGAN [46], instead of the diffusion model. We observe the results in Tab. 22. The results imply that using recently better-explored architecture yields better results when using the same transformations. This understanding strengthens our hypothesis for the robustness and efficiency of diffusion models.

Table 21: DiffTime scaling laws, KDD Cup

| #P | KDD Cup | | |
|---|---|---|---|
| | marg$\downarrow$ | class$\uparrow$ | pred$\downarrow$ |
| 3.4M | .026 | $.001 \pm .000$ | $1103.27 \pm 753.21$ |
| 13.5M | .022 | $.001 \pm .000$ | $902.77 \pm 632.74$ |
| 54M | .024 | $.000 \pm .000$ | $930.83 \pm 607.39$ |
| 104M | .024 | $.001 \pm .000$ | $970.21 \pm 693.19$ |
| 121M | .024 | $.000 \pm .000$ | $968.18 \pm 611.15$ |

Table 22: Other image generative models resutls

| Models/Datasets | KDD | | | NN Daily | | | Stocks | |
|---|---|---|---|---|---|---|---|---|
| | Marginal $\downarrow$ | Classifier $\uparrow$ | Predictor $\downarrow$ | Marginal $\downarrow$ | Classifier $\uparrow$ | Predictor $\downarrow$ | Disc $\downarrow$ | Pred $\downarrow$ |
| Style GAN | 0.020 | 0.001 | 0.233 | 0.020 | 0.091 | 2.100 | 0.276 | 0.042 |
| NVAE | 0.008 | 0.031 | 0.107 | 0.020 | 0.089 | 0.600 | 0.081 | 0.049 |

