# OpenReview forum: "Utilizing Image Transforms and Diffusion Models for  Generative Modeling of Short and Long Time Series"
_NeurIPS.cc/2024/Conference — NeurIPS 2024 poster_

### Official Review · Reviewer_DtfF · 2024-06-14

**Soundness:** 2
**Presentation:** 3
**Contribution:** 2
**Rating:** 5
**Confidence:** 4

**Summary:**

In this paper, authors propose to convert general time sequences into images by employing invertible transforms and incorporate advanced diffusion vision models to process short- and long-range time-series within the same framework. Through experiments, improvements have been made on multiple tasks, such as unconditional generation, interpolation and extrapolation.

**Strengths:**

1. The idea of converting time sequences to images is interesting.
2. Extensive experiments are conducted on several datasets of different tasks to evaluate the effectiveness of the proposed model. Besides, sufficient analysis and discussion make the results more convincible.
3. The overall organization of the paper is clear, and the writing is easy to follow.

**Weaknesses:**

1.  The time-series to image (ts2img) transform methods and diffusion backbone used in the paper are all from other existing works, which compromise the novelty and significance of the paper.
2. In Sec.5.1 and Sec.5.2, the ts2img transform methods used for short- and long-range generation are different. It is hard to tell whether the transform choice or the diffusion model plays a more important role. Taken together with Sec.5.5, the proposed method is only effective (i.e. outperforms others) with some specific transform method on different tasks, which conflicts with the authors’ claim that the proposed model can seamlessly process sequences with different lengths.

**Questions:**

Please see the weaknesses part above.

**Limitations:**

Yes.

---

> ### Author Rebuttal · Authors · 2024-08-06
>
> We appreciate the reviewer's recognition of the comprehensiveness of our experiments and the clarity of our writing. We also thank the reviewer for raising concerns and points that helped deepen the discussion. Below, we address these points and are more than happy to respond to any further concerns.
>
> > ***The time-series to image (ts2img) transform methods and diffusion backbone used in the paper are all from other existing works, which compromise the novelty and significance of the paper.***
>
> Our work considers the problem of generative modeling of varying-length (short-to-very-long) sequences. The significance of our paper corresponds to the high significance of this problem in the time series community. Thus, any new advances (including ours) on the front of generative modeling of varying-length sequences is *significant*. In particular, our paper is first to set a uniform strong baseline approach for generating time series of short-to-very-long lengths, where previous works fail to do so. Particularly, while simple and easy to implement, our method is robust and outperforms other methods across different length setups with lower computational complexity. Finally, we also show that enlarging the number of parameters is not enough to improve performance of previous works.
>
> The *innovation (novelty)* of our paper stems from our novel view of the problem as a visual task, and from our novel solution (illustrated in Fig. 1). To the best of our knowledge, these novel contributions were not suggested in the literature. Specifically, our approach bridges progress in generative diffusion vision research with time series generation through a novel combination and design of time series to image transforms and diffusion models, yielding a robust and innovative framework for generative time series modeling. We agree with the reviewer that some of the components we use (the diffusion model, the time series to image transforms) are not new. However, we do not believe that using established building blocks compromises novelty. Many papers suggest novel solutions to challenging problems based on existing building blocks, and our work aligns with this line of research.
>
>
> > ***In Sec.5.1 and Sec.5.2, the ts2img transform methods used for short- and long-range generation are different. It is hard to tell whether the transform choice or the diffusion model plays a more important role. Taken together with Sec.5.5, the proposed method is only effective (i.e. outperforms others) with some specific transform method on different tasks, which conflicts with the authors' claim that the proposed model can seamlessly process sequences with different lengths.***
>
> Thank you for highlighting this topic and allowing us to clarify. In Table 2 in the attached PDF, we directly compare the results of Delay Embedding (DE) with the second-best methods. Our results demonstrate that our model using DE significantly and consistently outperforms the second-best results (LS4 in the long setup and DiffTime in the short setup). In Sec. 5.5, we did not initially include a comparison to the second-best method per benchmark. However, including such comparison for long and short setups, makes it clear that our framework with DE is robust, outperforming the competition in 9 out of 10 comparisons, sometimes by as much as 50%. In the final revision, we plan to suggest users to first consider DE for generative modeling, before moving on to other time series to image transforms.
>
> Additionally, we would like to emphasize that the type of time series to image transform is set by a hyper-parameter in our framework. All modern learning algorithms are subject to certain hyper-parameters, and their performance varies depending on the specific hyper-parameter values. This dependence on hyper-parameters of ML algorithms is not considered typically as conflicting with consistency or stability claims. Finally, we also note that while our framework remains consistent across different length setups, LS4 fails completely in short generation, and DiffTime, as shown in our responses to reviews by SryL and iRua, fails in long sequence generative modeling. This further underscores the robustness of our framework with the DE transform to multiple lengths.

---

> > ### Comment · Reviewer_DtfF · 2024-08-08
> >
> > Thank you for the additional information. After reconsidering the contribution of this paper, I agree with your views on novelty. And the extra experiments help address my concerns. As you said, the type of time-series to image transform is set by a hyper-parameter, but I didn’t find the details about it. Instead of manual choice, it would be better if the transform type could be automatically chosen according to the time-series length. All things considered, I will increase my rating. Good luck to you.

---

> ### Author Response · Authors · 2024-08-11
> **Response**
>
> Thank you for your response and for reconsidering the evaluation of our work.
>
> We will clarify in the main paper that image transformation is selected as a hyperparameter and direct readers to Sections B.1, B.2, and B.3 of the appendix, where all relevant hyperparameter details, including image transformation, are described. We appreciate your feedback, which has improved the clarity of our work.
>
> Regarding the automatic choice based on sequence length, we will emphasize in the final revision that some transformations, like Delay Embedding, are robust across any length, while others, such as STFT, are better suited for specific lengths, like long or ultra-long sequences.
>
> We're happy to address any further questions you may have.

---

### Official Review · Reviewer_SryL · 2024-07-10

**Soundness:** 3
**Presentation:** 3
**Contribution:** 3
**Rating:** 6
**Confidence:** 3

**Summary:**

The paper argues for the use of image generative modelling architectures for the time-series generative modelling task. Doing so involves converting a time-series to an image-shaped object, modelling it as an image, and then converting back.

**Strengths:**

- This is a simple idea that is shown to work well in most of the experimental settings
- It allows the use of image architectures, which have been extensively investigated in the literature, for other domains
- The paper is mostly clearly written

**Weaknesses:**

Weaknesses:
- The authors mention that their approach "requires slightly higher computational resources". They go into more detail in Appendix C.10 but I would appreciate further detail. In particular, the cost is given in terms of hours/minutes, but I cannot see anything about e.g. GPUs used or FLOPs used. Are the number of GPUs matched between methods? A comparison in terms of GPU-hours for the same GPU type would be informative. To be really confident of the improvement, ideally there would be a comparison in which the methods were compared when given equal GPU-hours. Investigating how the performance of different methods scales as training FLOPs are increased would also be very interesting.
- It is a little unclear whether the advantage shown comes from the fact that image architectures are just better-explored than time-series architectures, or whether the inductive bias of transforming into an image shape is helpful. Can the authors comment on this? An interesting potential experiment would be to try training an image architecture with an older architecture and compare its performance.
- A more detailed description of the Predictive and Discriminative metrics used in Section 5.1 would be helpful.

**Questions:**

See weaknesses

**Limitations:**

Adequately addressed

---

> ### Author Rebuttal · Authors · 2024-08-06
>
> We are thankful to Reviewer SryL for generally identifying the simplicity of our approach and recognizing its ability to solve existing shortcomings and the extensive evaluation where we outperform baselines. We also would like to thank them for their observations, comments, and suggestions that helped deepen our discussion and improve the paper. Below, we address the reviewer's concerns. Given the opportunity, we will incorporate the responses below into the final revision.
>
> > ***... is given in terms of hours/minutes, but I cannot see anything about e.g. GPUs used or FLOPs used. Are the number of GPUs matched between methods?***
>
> Yes, the comparison is done in exactly the same environment. The software environments we use are CentOS Linux 7 (Core) and PYTHON 3.9.16, and the hardware is NVIDIA RTX 3090. In addition, all experiments run on a single GPU for all methods.
>
> > ***... Investigating how the performance of different methods scales as training FLOPs are increased would also be very interesting.***
>
> We analyze the FLOPs used per method and show the results in Table 7. We observe that our method is the most efficient FLOP-wise across all sequence lengths. We could not run LS4 on KDD Cup with $>100$M parameters, and thus, we omit the FLOPs computation for $100$M and $150$M set-ups.
>
>
> > ***...It is a little unclear whether the advantage shown comes from the fact that image architectures are just better-explored than time-series architectures, or whether the inductive bias of transforming into an image shape is helpful...***
>
> We thank the reviewer for pointing this out. Our goal in this paper is to leverage recent advancements in computer vision to develop an elegant and robust solution for time-series data, addressing different sequence lengths and setting a baseline for handling short, long, and ultra-long sequences. We aim to take advantage of the fact that image architectures are more thoroughly explored. We hypothesize that the improvements we observe are largely due to the more advanced development of image architectures compared to time-series architectures. To further investigate this and following the reviewer's suggestion, we used NVAE [2], and StyleGAN [3], instead of the diffusion model. We observe the following results:
>
>
> | Model       | Dataset   | Marginal ↓ | Classifier ↑ | Predictor ↓ | Disc ↓ | Pred ↓ |
> |-------------|-----------|------------|--------------|-------------|------------|------------|
> | **Style GAN** | KDD       | 0.02       | 0.001        | 0.233       | -      | -    |
> |             | NN Daily  | 0.02       | 0.091        | 2.1         |    -      |   -      |
> |             | Stocks    | -          | -            | -           |   0.276  | 0.042  |
> | **NVAE**     | KDD       | 0.008      | 0.031        | 0.107       |   -     |   -     |
> |             | NN Daily  | 0.02       | 0.089        | 0.6         |   -    |   -    |
> |             | Stocks    | -          | -            | -           | 0.081  | 0.049  |
>
>
> The results imply that using recently better-explored architecture yields better results when using the same transformations. This understanding strengthens our hypothesis for the robustness and efficiency of diffusion models.
>
>
> > ***...A more detailed description of the Predictive and Discriminative metrics...***
>
> We used the benchmark proposed in [1] to evaluate short-term unconditional generation, and we adhere to its evaluation protocol. This protocol comprises two scores: a predictive score and a discriminative score. Discriminative score: given the original data labeled as 'true' and generated data labeled as 'fake', we split the data into $80$%, $20$% for train and test data, respectively. Then, We train a model to discriminate between the samples. Finally, we report $|0.5 - \text{pred}|$ where pred is the models' accuracy over the test set. The lower the score, the better is the generated data since the model struggles to discriminate between the two. Predictive score (train-on-fake-test-on-real): given the generated data, we train a sequence-prediction model to predict next-step temporal vectors over each generated input sequence. Finally, we evaluate the trained model on the original data. Performance is measured with the mean absolute error (MAE). Importantly, these metrics are used in most of the time series generation papers for short sequences.
>
>
> The rest of our tables are presented in the attached additional PDF file.
>
>
>
>
>
> [1] Time-series Generative Adversarial Networks by Yoon et al.
>
> [2] NVAE: A Deep Hierarchical Variational Autoencoder by Arash Vahdat, Jan Kautz.
>
> [3] A Style-Based Generator Architecture for Generative Adversarial Networks by Karras et al.

---

> > ### Comment · Reviewer_SryL · 2024-08-13
> >
> > Thank you for your thorough response to my concerns. Your comment that your aim is to "leverage recent advancements in computer vision to develop an elegant and robust solution for time-series data" does help to clarify the contribution of this paper for me. I have raised my score to a 6.

---

### Official Review · Reviewer_iRua · 2024-07-12

**Soundness:** 3
**Presentation:** 3
**Contribution:** 3
**Rating:** 6
**Confidence:** 3

**Summary:**

The paper proposes using invertible transforms to map varying-length time series to images. Using this technique, generative modeling of time series can be done using diffusion vision models. The authors demonstrate state-of-the-art performance on unconditional generation, interpolation, and extrapolation on short and long time series benchmarks. An additional contribution of the paper is the introduction of a novel benchmark for ultra-long (>10k timesteps) time series.

**Strengths:**

- The main idea (transforming time series to images to exploit existing image-generation methods) is quite elegant and investigated well.
- The experiments are thorough: unconditional and conditional generation is evaluated on short and long time series benchmarks. Furthermore, the results are convincing.

**Weaknesses:**

- One crucial point is that the proposed method involves 1-2 orders of magnitude more parameters than e.g. LS4, a close competitor (Table 16). I think this point should be more clearly emphasized and investigated in the main paper.
  - The authors mention that in terms of wall-clock time, their method's training and inference efficiency is comparable despite the difference in size. However, this seems to be mostly a statement on how much work has been done improving efficiency of image diffusion models -- it may be possible to drastically improve efficiency of existing time series methods.

**Questions:**

- How does the number of parameters compare between the proposed method and all the other methods you compare to, e.g. in Tables 1 and 2?
- Were any experiments done evaluating how the proposed method compares to e.g. a parameter-matched LS4 model?
- Similarly, were any scaling law experiments done to investigate how the performance of the proposed method improves as the size of the image diffusion model increases?
  - For example, it would be quite interesting to find that the scaling of the image diffusion model architecture is better than the scaling of e.g. LS4 or time series diffusion architectures.

**Limitations:**

The authors adequately address the limitations of their work.

---

> ### Author Rebuttal · Authors · 2024-08-06
>
> We are thankful to Reviewer iRua for generally identifying the elegancy of our approach and its generalizability. We also would like to thank them for their observations, comments, and suggestions that helped deepen our discussion and improve the paper. Below, we address the reviewer's concerns. Given the opportunity, we would be happy to incorporate the reviewer's concerns into a final revision.
>
>
> > ***One crucial point is that the proposed method involves 1-2 orders of magnitude more parameters than e.g. LS4, a close competitor...***
>
> Thank you for raising this question. Our original submission included models with 1-2 orders of magnitude more parameters vs. LS4. However, following the reviewer's questions and suggestions, we performed a thorough scaling study. We find that our approach still achieves SOTA results on short sequences with one order of magnitude less parameters in comparison to LS4. We attain SOTA results on longer sequences with models that are on par to LS4 in size. Our analysis is provided in Table 3 and Table 4 for the DiffTime method, Table 5 for LS4, and Table 6 for our method. In addition, scaling law results for each method are in Fig. 1 in the attached additional PDF file.
>
>
> > ***... how much work has been done improving efficiency of image diffusion models -- it may be possible to drastically improve efficiency of existing time series methods.***
>
> The primary motivation in our work was to leverage recent advancements in computer vision to enhance time-series data analysis. Our design is such that it can directly utilize the work done to improve diffusion models, as well as future extensions. In contrast, while it is possible to drastically improve time series methods, it is a major unknown. At the very basic, our approach uses convolutions whose computation is supported in GPU hardware, whereas time series techniques are mostly based on the transformer, which is not yet implemented in hardware.
>
> > ***...number of parameters compare between the proposed method and all the other methods...***
>
>
> We have extended Tab. 16 from the App. C.10 and included the updated table in Table 1. Specifically, we provide a comparison with the second-best methods for short sequences, DiffTime and GT-GAN. However, we emphasize that these methods do not scale well for ultra-long sequences (DiffTime) and even for long sequences in the case of GT-GAN, whose time complexity depends on the sequence length.
>
> > ***... evaluating how the proposed method compares to, e.g., a parameter-matched LS4 model...***
>
> > ***... scaling law experiments done to investigate how the performance of the proposed method improves as the size of the image diffusion model increases...***
>
>
> We extend our evaluation by increasing the parameters of LS4 and DiffTime while decreasing the parameters of our model (Tables 3,4,5,6). This evaluation includes three datasets: Stocks (short), nn5daily (long), and KDD Cup (ultra-long). Evidently, increasing the number of parameters does not improve model performance. We believe this is a significant contribution of our paper, as it demonstrates that simply enlarging previous methods does not necessarily enhance results. Moreover, in the case of LS4 on the KDD Cup dataset, increasing the model's parameters to 100 million results in memory collapse, making it infeasible to run a batch size of one with the current resources used for training all models.
>
> Finally, we present our results in Figure 1. We appreciate the reviewer for highlighting this issue and would like to include these results in our revision.
>
> The figure and the tables are presented in the attached additional PDF file.

---

> > ### Comment · Reviewer_iRua · 2024-08-11
> >
> > Thanks to the authors for the response! The new results (regarding matched parameter / FLOPs comparisons) are enlightening and compelling. I will raise my score accordingly.

---

> > > ### Author Response · Authors · 2024-08-12
> > > **Response**
> > >
> > > Thank you for your response and for reevaluating our work. We are happy that our response address all your concerns and questions, and we would be more than willing to address any additional ones you may have.

---

### Author Rebuttal · Authors · 2024-08-06

We have attached the PDF. References from the rebuttal comments are directed to the tables or figures in this PDF.

---

### Decision · Program_Chairs · 2024-09-25

**Decision:**

Accept (poster)

**Comment:**

This work presents a simple but effective approach for time-series modeling by transforming the timeseries into an image and then applying standard image diffusion modeling. While the novelty of the approach is relatively limited (and is similar to prior work applying diffusion to spectrograms), the authors demonstrate consistent empirical performance across standard timeseries benchmarks and show that their approach improves over prior work like LS4.